# Interferon-Mediated Long Non-Coding RNA Response in Macrophages in the Context of HIV

**DOI:** 10.3390/ijms21207741

**Published:** 2020-10-19

**Authors:** Tinus Schynkel, Matthew A. Szaniawski, Adam M. Spivak, Alberto Bosque, Vicente Planelles, Linos Vandekerckhove, Wim Trypsteen

**Affiliations:** 1HIV Cure Research Center, Department of Internal Medicine and Pediatrics, Ghent University and Ghent University Hospital, 9000 Ghent, Belgium; Tinus.Schynkel@UGent.be (T.S.); Linos.Vandekerckhove@UGent.be (L.V.); 2Division of Microbiology and Immunology, Department of Pathology, University of Utah School of Medicine, Salt Lake City, UT 84132, USA; matt.szaniawski@hsc.utah.edu (M.A.S.); vicente.planelles@path.utah.edu (V.P.); 3Division of Infectious Diseases, Department of Medicine, University of Utah School of Medicine, Salt Lake City, UT 84132, USA; adam.spivak@hsc.utah.edu; 4Department of Microbiology, Immunology and Tropical Medicine, School of Medicine & Health Sciences, George Washington University, Washington, DC 20037, USA; abosque@email.gwu.edu

**Keywords:** long non-coding RNA, interferon, Human Immunodeficiency Virus (HIV), macrophage, RNA-seq

## Abstract

Interferons play a critical role in the innate immune response against a variety of pathogens, such as HIV-1. Recent studies have shown that long non-coding genes are part of a reciprocal feedforward/feedback relationship with interferon expression. They presumably contribute to the cell type specificity of the interferon response, such as the phenotypic and functional transition of macrophages throughout the immune response. However, no comprehensive understanding exists today about the IFN–lncRNA interplay in macrophages, also a sanctuary for latent HIV-1. Therefore, we completed a poly-A+ RNAseq analysis on monocyte-derived macrophages (MDMs) treated with members of all three types of IFNs (IFN-α, IFN-ε, IFN-γ or IFN-λ) and on macrophages infected with HIV-1, revealing an extensive non-coding IFN and/or HIV-1 response. Moreover, co-expression correlation with mRNAs was used to identify important (long) non-coding hub genes within IFN- or HIV-1-associated gene clusters. This study identified and prioritized IFN related hub lncRNAs for further functional validation.

## 1. Introduction

The innate immune response is an essential line of defense against viruses and other microbial pathogens. A vital part of the innate immunity depends on interferons (IFNs), cytokines that are released to enhance the anti-viral state of neighboring cells in an attempt to stop the spreading of the infection [1]. The initiation of a typical IFN-mediated response starts with the recognition of pathogen-associated molecular patterns (PAMPs) by Toll-like, RIG-I or MDA5 receptors, which will trigger the expression of IFN genes [1,2]. Based on their receptor, IFNs are classified in three classes. Type I IFNs consist of 12 IFN-α subtypes, IFN-β, IFN-ε, IFN-κ and IFN-ω, while IFN-γ is the only Type II IFN in humans. The more recently discovered Type III IFNs comprise four IFN-λ molecules [1,3]. IFN-receptor binding triggers an extensive signaling cascade, including the JAK-STAT pathway, that will induce the expression of a broad range of antiviral and antimicrobial genes [4].

Next-generation sequencing has revealed that of all genes transcribed, only about 2% code for proteins [5]. For a long time, the large fraction of non-coding genes was regarded as transcriptional noise. However, over the last years, it has become clear that numerous classes of non-coding RNAs (ncRNAs) play central roles within cell biology [6]. In addition to rRNAs, tRNAs and microRNAs, the class of long non-coding RNAs (lncRNAs) has proven to possess unique characteristics. They are somewhat arbitrarily defined as ncRNAs with a length of more than 200 nucleotides. As with mRNAs, their transcription depends most often on RNA polymerase II activity and as often as not they are spliced and/or polyadenylated [7]. A significant fraction of lncRNAs are less conserved than protein-coding genes and are also expressed at lower levels compared to protein-coding genes, often specific to tissues, cell types or (disease) states [7,8]. As a result of their protein-, DNA- and RNA-binding capacities, lncRNAs have nonetheless demonstrated a broad functional repertoire, including chromatin modification and (post-) transcriptional regulation [6,9]. 

In an IFN context, lncRNAs are involved in a reciprocal feedforward/feedback relationship with IFN expression [10], likely contributing to the cell type-specificity of the IFN response [11]. One of the cell types affected by the IFN–lncRNA interplay are macrophages, cells of the myeloid lineage that occupy key roles in innate and adaptive immunity, tissue homeostasis, development and repair. LncRNAs play important roles in macrophage polarization [12,13] and cytokine-driven immune activity [14]. Complete insight into the IFN–lncRNA interplay in macrophages would provide a better understanding of the regulation of the immune system, aiding research on immune diseases and infections, such as the human immunodeficiency virus type I (HIV-1). IFNs have shown to be a potent barrier for HIV-1 infection [15,16] and growing evidence suggests that macrophages constitute a long-lived HIV-1 reservoir, hampering the search for a definite HIV-1 cure [17,18]. 

To our knowledge, no comprehensive analysis of the (long) non-coding RNA/interferon response in macrophages has been performed. Therefore, we completed a poly-A+ RNAseq analysis on monocyte-derived macrophages (MDMs) treated with members of all three types of IFNs (IFN-α, IFN-ε, IFN-γ or IFN-λ) and on macrophages infected with HIV-1, revealing an extensive non-coding IFN and/or HIV-1 response. Moreover, co-expression correlation with mRNAs was used to identify important (long) non-coding hub genes within IFN- or HIV-1-associated gene clusters. 

## 2. Results

### 2.1. Transcriptome of Macrophages Stimulated with Interferons or Subjected to HIV-1 Infection

RNA sequencing data available from a previous study [19] were mined to uncover effects of IFN stimulation or HIV infection on the non-coding transcriptome of MDMs (NCBI, accession No. GSE158434). Szaniawski et al. induced MDMs for 18 h with a single interferon (IFN-α, IFN-ε, IFN-γ or IFN-λ) or infected MDMs for 6 h with HIV-1. Subsequent transcriptome analysis via RNA-sequencing (RNA-seq) was performed. In total, 407 million reads were mapped to the human GRCh38 genome and for every sample the proportion of uniquely mapped reads exceeded 87%.

IFN-ε stimulation provoked the most extensive differential expression profile compared to the non-treated control with a total of 3614 differentially expressed genes (|log_2_ fold change| > 1; Padj < 0.05), of which 2892 were mRNAs, 187 pseudogenes and 493 lncRNAs. IFN-α, IFN-γ and IFN-λ caused differential expression of 2147, 2568 and 683 genes, respectively (Figure 1A). Almost as many genes are downregulated as upregulated. Interestingly, 36 non-coding genes were differentially expressed in all four IFN stimulated conditions (Figure 1B), presumably as part of a broad interferon response, while other non-coding genes were induced in an interferon type-specific manner. Upon HIV-1 infection, 474 genes were differentially expressed, of which 352 were mRNAs, 38 pseudogenes and 78 lncRNAs (Figure 1A). Of the non-coding fraction, seven differentially expressed genes show overlap with the broad interferon response genes: lncRNAs NRIR, MIR3945HG, C8orf3, AC053503.1 and AL359551.1 and pseudogenes GBP1P1 and NCF1C (Appendix A). LncRNAs NEAT1, HEAL, GAS5 and MALAT1 have all been previously reported to be associated with HIV infection, but we observed no significant differential expression in HIV-1 infected MDMs. The top 25 differentially expressed non-coding genes for all conditions are displayed in a heatmap (Figure 1C).

To validate the RNAseq data, a qPCR quantification was performed on three protein coding genes and five non-coding genes (Figure 2). For protein coding genes MX1, IFIT2 and RSAD2/viperin, the qPCR analysis is in line with the previously obtained RNAseq results. Only in the HIV-1 infected condition, the qPCR results deviate from the RSAD2/viperin log_2_ fold change observed with RNAseq. The fold change of the non-coding genes NRIR, FIRRE and RP11-177F15.1 observed with qPCR is also in agreement with the RNAseq analysis. DANCR and RP3-477O4.14 have aberrant qPCR results from the RNAseq data in one or two conditions. 

### 2.2. Gene Enrichment Analysis

Toppgene analysis [20] on the differentially expressed genes in every condition was performed to determine mRNA enrichment in Gene Ontology Biological Process (GO_BP) terms [21] and Kyoto Encyclopedia of Genes and Genomes (KEGG) pathways [22]. The top five enriched GO_BP terms and KEGG pathways for every condition are displayed in Appendix A. Both interferon stimulation and HIV-1 infection highly enrich GO_BP terms associated with cytokine response and signaling, as well as the response to external biotic stimuli such as viruses, supporting the value of this RNA-seq dataset to screen for IFN related lncRNAs in the context of macrophages. In line with the GO_BP terms, all conditions are enriched for pathways associated with cytokine signaling (e.g., cytokine–cytokine receptor interaction, chemokine signaling pathway and NOD-like receptor signaling pathway) and infections such as Influenza A virus, Herpes simplex or *Staphylococcus aureus*. 

### 2.3. Weighted Gene Co-Expression Network Analysis (WGCNA) to Identify IFN Related lncRNA Hub Genes

Most non-coding genes are still unannotated and functionally unexplored. One way to screen for functionally relevant non-coding genes is to look at co-expression with known mRNAs. A WGCNA network was constructed that hierarchically clusters all genes, based on a bi-weight mid-correlation of RNA expression patterns over all conditions [23]. Many lncRNAs have a repressive impact on their mRNA binding partners and thus an opposite expression profile [6]. Therefore, an unsigned network was used that allows genes with opposing expression patterns to cluster together. By dynamically cutting the hierarchical dendrogram, 16 modules (clusters) of correlated genes was detected (Figure 3A). Toppgene analysis was used to find biologically relevant modules that are enriched for genes involved in interferon or antiviral response (Table 1 and Table 2). The brown module, containing 2870 mRNAs and 769 non-coding RNAs, is among others highly enriched in the GO_BP terms viral process, response to interferon-gamma and innate immune response, as well as the pathways interferon signaling, herpes simplex infection and adaptive immune system. Hence, non-coding genes within this module have a high likelihood of contributing to similar biological processes. Correlation of the brown module’s eigengene with the interferon and HIV-1 infection trait status shows a significant positive correlation (*p* < 0.05) with Type I interferon induction (IFN-α and IFN-ε), while having a negative correlation with HIV-1 infection (Figure 3B). No significant trait–module correlation was seen for the brown module with IFN-γ or IFN-λ. Based on module membership (MM: the correlation between an individual gene and the module eigengene) and gene significance (GS: the correlation between the gene and the trait), intramodular hub genes for the brown module were identified (GS > 0.5; MM > 0.9). These hub genes represent highly interconnected genes with potential key roles within pathways correlated with the particular traits. For IFN-α and IFN-ε stimulation and HIV-1 infection, respectively, 368, 353 and 387 hub genes were identified, of which 20, 22 and 21, respectively, are non-coding genes (Figure 3C, Appendix A).

### 2.4. Interaction Network Construction of WGCNA Hub Genes in IFNα and IFNε Stimulated Conditions

To further explore the role and identify major IFN related non-coding hub genes, a poly-A+ ncRNA–mRNA–protein interaction network was constructed for hub genes in the brown WGCNA module which were also found differentially expressed in IFN-induced conditions. The network consists of protein–protein interactions and lncRNA–mRNA interactions that were predicted based on the genomic location of the genes and the normalized binding free energy between the lncRNA and mRNA sequence.

Figure 4A shows the network for 128 differentially expressed, IFN-α correlated hub genes, including 15 non-coding genes. LncRNAs TNK2-AS1 and FIRRE display a high interconnectivity with 23 and 5 predicted lncRNA–mRNA interactions. The mRNA of IFITM1, a protein with an immune response signaling function, shows very high binding capacity with 15 non-coding hub genes. 

In the second network, constructed with 115 differentially expressed, IFN-ε correlated hub genes, the onco-lncRNA DANCR stands out with 21 predicted lncRNA–mRNA interactions (Figure 4B). The lncRNAs RP3-477O4.14 and AC064834.3 are predicted to interact with six and four mRNAs, respectively. 

### 2.5. Interaction Network Construction of Differentially Expressed HIV-1 Correlated Genes

A network was constructed with the (non-hub) genes that were differentially expressed upon HIV-1 infection (Figure 5). Two pseudogenes of CD24 (CD24P1 and CD24P2) exhibit a central role in this network with 18 and 31 predicted interactions with mRNAs. In addition, the lncRNAs “Negative regulator of interferon” (NRIR) and RP11-231G3.1 are predicted to interact with seven and four mRNAs, respectively.

## 3. Discussion

In the present study, we characterized the transcriptional response induced by IFNs in monocyte-derived macrophages. Macrophages are phagocytic cells that have a very diverse set of functions, including pattern recognition of foreign ligands, antigen processing and presentation, release of inflammatory mediators, tissue homeostasis and repair [27]. To be able to fulfill all these functions, macrophages have a strong phenotypic plasticity [28,29], and, based on stimuli from their microenvironment, they can polarize into different types (classically activated M1 or alternatively activated M2 macrophages) [28,30]. Cytokines such as IFNs have an important role mediating the functional transition of macrophages throughout the innate immune response [31]. Wentker et al. constructed a map of inflammation-related signal transduction pathways in macrophages, illustrating the overall dynamic coordination by transcriptional and post-transcriptional gene regulation [32] and Hu et al. showed an extensive transcriptional response to the cytokine interleukin-27 in macrophages [14]. In this study, for members of all three IFN subclasses (IFN-α and IFN-ε for Type I IFNs, IFN-γ for Type II IFNs and IFN-λ for Type III IFNs), the expression profile was examined, resulting in the identification of 2147 (IFN-α), 3615 (IFN-ε), 2568 (IFN- γ) and 683 (IFN- λ) differentially expressed genes. Gene enrichment analysis demonstrates that these genes are unsurprisingly enriched for GO_BP terms and KEGG pathways that are associated with cytokine response and signaling (e.g., cytokine–cytokine receptor interaction, chemokine signaling pathway and NOD-like receptor signaling pathway) and associated with the response to external biotic factors (e.g., Influenza A virus, Herpes simplex or *Staphylococcus aureus*). This validates the fact that the transcriptome examined in the present study is at least for a considerable part provoked by IFN stimulation and that the IFN stimulation triggers a transcriptomic response against infections of diverse origin.

On average, 20%, or a total of 1082 discovered unique differentially expressed genes over all four IFN conditions in this study, is categorized as non-coding. Thus, their main function does not depend on mRNA to protein translation, but it is rather RNA transcription itself or the RNA molecule that fulfills the purpose of the gene. Many of them are lncRNAs, but differentially expressed pseudogenes are present too. Pseudogenes are DNA segments that are in general considered to be non-functional relics derived from unfaithful gene duplications or retrotransposition of processed mRNAs [33]. However, several studies have shown that pseudogene expressed non-coding RNAs can have a functional role, such as the regulation of their protein-coding counterparts [34]. Therefore, pseudogenes were included in this analysis. We note that in the current study we analyzed poly-A purified RNA-seq data. As non-coding genes such as lncRNAs are only in approximately 50% of the cases poly-adenylated [7], the number of discovered non-coding differentially expressed genes is most likely an underestimation. Nevertheless, to our knowledge this is the first comprehensive analysis of the (long) non-coding interferon response in macrophages for all three interferon types. 

IFN-α and IFN-ε (type-I IFNs) stimulation provoked differential expression of 429 and 723 non-coding genes, respectively, of which about 50% of the genes was upregulated. This is in line with the RNA-seq analysis done by Carnero et al. of HuH7 cells treated for 72 h with 10,000 units/mL of IFNα2. They reported differential expression of 890 putative non-coding genes, with also half of the genes upregulated [35]. We demonstrate that IFN-γ stimulation induces the differential expression of 474 non-coding genes in macrophages. IFN-γ is a known macrophage activator that induces a pro-inflammatory phenotype M1 [36]. Huang et al. differentiated macrophages into M1 cells using IFN-γ and LPS and utilized a lncRNA-specific microarray to identify 9343 lncRNAs with a twofold differential expression compared to unactivated macrophages. The IFN-λ response in this study is the least extensive, with only 683 (of which 142 are non-coding) differentially expressed genes. Read et al. however demonstrated that macrophages are primary responders to IFN-λ, gaining IFN-λ receptor expression upon differentiation from CD14+ monocytes. Based on the induction of the interferon stimulated genes (ISGs) RSAD2/viperin and ISG15, they concluded that IFN-γ activates the transcription of ISGs [31]. However, we show here that, although there is measurable activation of ISGs as a result of IFN-λ stimulation, this effect is less pronounced than that observed for IFN Type I and II stimulation. 

Next to IFN stimulation, we also assessed the transcriptomic change of macrophages following HIV-1 infection. Although the main targets for HIV-1 infection are CD4+ T-cells, considerable evidence suggests that HIV-1 also infects tissue macrophages, which can give rise to a long-lived reservoir under antiretroviral therapy (ART) [17,18,37,38]. In vitro IFNs have been shown to be a strong barrier for HIV-1 infection during the early stages of the HIV-1 life cycle. Goujon and Malim reported that IFN-α blocks HIV-1 infection in macrophages, possibly in an ubiquitin-proteasomal correlated manner [16]. Tasker et al. showed that IFN-ε induces a potent anti-HIV-1 state by enhancing phagocytosis and the production of ROS [15]. Both of the previous studies mention that IFN blocks HIV-1 infection in macrophages to a significantly larger extend than in CD4+ T-cells, pointing at a dramatic difference in the IFN response between the two cells. On the other hand, Harman et al. showed that macrophages infected with HIV-1 fail to produce Type I and Type III IFNs, because of Vpr- and Vif-mediated inhibition of TBK1, a protein that indirectly activates NF-κB [39]. We report the differential expression of 474 genes in macrophages after 6 h of a lab-strain HIV-1 infection. In total, 112 genes or about 24% of them are categorized as non-coding, of which 59 are also differentially expressed in one or more IFN conditions. 

Interestingly, seven differentially expressed genes show overlap with all four IFN conditions: lncRNAs NRIR, MIR3945HG, C8orf3, AC053503.1 and AL359551.1 and pseudogenes GBP1P1 and NCF1C. Negative regulator of interferon response (NRIR) is a known lncRNA that was first described by Kambara et al. [40] as a lncRNA that was ~100-fold upregulated upon IFN-α treatment of hepatocytes. NRIR knockdown reduced HCV infection in IFN stimulated hepatocytes and resulted in upregulation of several ISGs, including two ISGs neighboring its locus, CMPK2 and RSAD2/viperin. This indicated that NRIR is involved in a negative IFN feedback mechanism. Here we show that NRIR transcription is induced in macrophages by all three types of IFNs and also by HIV-1 infection, indicating that NRIR possibly has a negative effect on the IFN response against HIV in macrophages. GBP1P1 was first identified as an interferon Type I induced gene in HuH7 cells by Camero et al. [41]. They reported that GBP1P1 mimics the expression of the ISG GBP1 and that GBP1P1 is induced by the influenza virus. Furthermore, GBP1P1 is upregulated in liver cells of HCV patients and in chronically infected HIV patients. In this study we found that GBP1P1 is induced in macrophages by every IFN-type (up to ~350 fold) and by HIV-1 infection (~3 fold). MIR3945HG is a lncRNA that was previously found to be differentially expressed in macrophages infected with *Mycobacterium tuberculosis* [42]. Transcription of MIR3945HG might also regulate the expression of its hosting miRNA, rather than function directly itself. Further research should investigate what molecular effects, if any, lncRNAs and pseudogenes such as MIR3945HG and GBP1P1 have on HIV infection.

The RNA-seq data were validated by performing a qPCR on a selected set of genes. In general, the qPCR outcome was in line with the RNA-seq data, although a few anomalies were observed. These discrepancies could partially be explained by the general low read counts of lncRNAs. Hence, a subtle difference in counts could have a disproportionate qPCR outcome. Additionally, the different way of normalization, multiple splicing variant detection and the pooling of the RNAseq triplicates for qPCR analysis may explain these few discrepancies.

Based on the WGCNA analysis and subsequent gene enrichment analysis, an IFN associated module was identified that positively correlates with IFN Type I induction and negatively with HIV-1 infection. Networks constructed with predicted interactions revealed key non-coding genes within this module. We want to note that the lncRNA–mRNA interactions are merely predictions that are in general not considered as a very common functional mechanism for lncRNAs. Therefore, future in vitro validation will have to assess actual lncRNA–mRNA binding and its functional impact. However, lncRNAs TNK2-AS1 and FIRRE are upregulated upon IFN-α stimulation and highly interconnected within the network. TNK2-AS1 has been identified by Wang et al. as an oncogenic lncRNA that is in a positive feedback loop with STAT3, which enhances angiogenesis in non-small cell lung cancer [43]. Fire Intergenic Repeating RNA Element (FIRRE) was reported by Lu et al. to positively regulate the expression of several inflammatory genes in response to LPS stimulation in macrophages [44]. Zang et al. showed that FIRRE is in a positive feedback loop with NF-κB and promotes the transcription of NLFP3 inflammasome through its interaction with hnRNPU [12,45].

In the IFN-ε network, we identified the lncRNAs DANCR, RP3-477O4.14 and AC064834.3 as key molecules within the IFN-ε hub. Differentiation antagonizing non-protein coding RNA (DANCR) is a known onco-lncRNA that works as a promotor for vital components of the oncogene network by sponging certain microRNAs and interaction with various regulating proteins [46]. Here, we show that DANCR is downregulated by IFN-ε stimulation and we similarly predict multiple DANCR-protein interactions, indicating a possible negative regulating function within the network. 

Within the HIV-1 network, NRIR is the best characterized key non-coding gene identified. In addition, two pseudogenes of CD24 (CD24P1 and CD24P2) have highly interconnected characteristics in the network. These pseudogenes were undetectable in all condition but the HIV-1 infected condition, pointing at a possible biomarker function for these pseudogenes. We note that in the HIV-1 infected condition a HIV lab strain was used that was tagged with a murine CD24 reporter gene. However, alignment of this murine CD24 sequence with the sequence human CD24P1 and CD24P2 genes showed plenty of mismatches, indicating that this is not the source of the CD24 pseudogene reads detected.

In summary, the innate immunity can act as a double-edged sword, acting as a first line of defense against diverse pathogens, but often bringing along side-effects that cause more harm than good. Understanding the transcriptional control of the innate immunity can aid us in the research to reshape the innate immunity in case of infection or immune diseases. Therefore, we identified and prioritized IFN-related hub lncRNAs for further functional validation.

## 4. Materials and Methods

### 4.1. RNA Sequencing Data Generation

RNA sequencing data were generated by Szaniawski et al. [19]. The sequencing data are publicly available at the National Center of Biotechnology Information under accession no. GSE158434.

Briefly, CD14+ monocytes from healthy donors were cultured for 7 days to allow differentiation to MDM. MDM were stimulated in triplicate with 25 ng/mL of a single interferon (IFN-α, IFN-ε, IFN-γ or IFN-λ) or infected with 250 ng HIV-1-BAL-HSA virus. Total RNA was isolated 18 h following interferon stimulation or 6 h post infection using the RNeasy minikit (Qiagen, Hilden, Germany). Intact poly(A) RNA was purified and stranded mRNA sequencing libraries were prepared using the Illumina (San Diego, CA, USA) TruSeq Stranded mRNA Library Preparation kit (catalog No. RS-122-2101 and RS-122-2102). On an Illumina HiSeq 2500 instrument (HCSv2.2.38 and RTA v1.18.61), a 50-cycle single-read sequence run was performed using HiSeq SBS kit v4 sequencing reagents (catalog no. FC-401-4002). 

### 4.2. RNA Sequencing Data Processing and Differential Gene Expression Analysis

The human GRCh38 genome and gene feature files were downloaded from Ensembl release 87 and a reference database was created using STAR version 2.5.2b with splice junctions optimized for 50 base pair reads [47]. Reads were trimmed of adapters and aligned to the reference database using STAR in two pass mode to output a BAM file sorted by coordinates. Mapped reads were assigned to annotated genes using featureCounts version 1.5.1 [48]. The output files from FastQC, Picard CollectRnaSeqMetrics, STAR and featureCounts were summarized using MultiQC [49] to check for any sample outliers. DESeq2 version 1.16 [50] was used and genes were considered differentially expressed when |log_2_ fold change| > 1 compared to the untreated control and Padj < 0.05. Genes were assigned to a gene biotype using the bioMart package (v2.26.1) in R (v3.5.2). Genes in the biotypes protein_coding, non_stop_decay, nonsense_mediated_decay, Artifact, TEC, IG_C_gene, IG_D_gene, IG_J_gene, IG_LV_gene, IG_V_gene, TR_C_gene, TR_J_gene, TR_V_gene, TR_D_gene and LRG_gene were considered protein coding genes. Non-coding genes were classified as antisense lncRNAs, intergenic lncRNAs, sense-overlapping lncRNAs and intronic lncRNAs; pseudogenes; and other non-coding genes (miRNA, misc_RNA, processed_transcript, snoRNA, rRNA, snRNA, Mt_tRNA, Mt_rRNA and scaRNA).

### 4.3. Gene Ontology and Pathway Enrichment Analysis

The ToppGene (ToppFun) webtool [20] was used to determine if the differentially expressed genes in each condition were enriched for Gene Ontology Biological Process (GO_BP) terms [21] or Kyoto Encyclopedia of Genes and Genomes (KEGG) pathways [22]. A cutoff of FDR B&H < 0.05 was used. The reported adjusted *p*-value is Bonferroni corrected. 

### 4.4. Weighted Gene Co-Expression Network Analysis (WGCNA)

A weighted gene co-expression network analysis [23] (WGCNA) (v1.69) was performed in R (v3.5.2) on the log_2_(x + 1) normalized count data. All genes were included to assure scale-free topology. An unsigned, single-block gene co-expression network was created based on bi-weight mid-correlation between genes with a soft threshold of power β = 6. The correlated genes were hierarchically clustered in a dendrogram. Modules were detected by Dynamic Tree Cutting, and, subsequently, modules with highly correlated eigengenes were merged (merge height = 0.25). The eigengene values of the modules were correlated with the external binary trait information (interferon stimulation; HIV-1 infection) to create a module–trait heatmap. Gene significance (GS) for interferon stimulation or HIV-1 infection was determined and hub genes within the modules were identified (GS > 0.2 and module membership MM > 0.8). Each module was checked for GO_BP term or pathway (KEGG/REACTOME) enrichment with the ToppGene webtool [20].

### 4.5. Molecular Interaction Analysis (STRING; LncTAR; Proximity)

Interaction networks were built in Cytoscape (v3.8.0). For the sake of comprehensibility, the number of nodes was reduced by only including genes with a minimal total count of 180 transcripts over all conditions, differential expression (|log_2_ fold change| > 1; Padj < 0.05) and stringent hub gene characteristics (|GS| > 0.5; |MM| > 0.9). The STRING (v11.0) webtool [24] was used to identify protein–protein interactions with a minimum interaction score of 0.4. The LncTar command line tool [25] was used to find mRNA–lncRNA interactions with a cut-off for normalized binding free energy (ndG) of -0.15 (high confidence). mRNAs in the genomic proximity of lncRNAs (50 kbp up- or downstream) were found by comparing transcript start and stop sites in R (v3.5.2) based on the Ensembl database. 

### 4.6. Relative Quantification by qPCR for RNAseq Validation

For qPCR validation, the RNA from the three replicates for every condition was pooled. Reverse transcription to cDNA was performed with the qScript cDNA Synthesis Kit (Quantabio, Beverly, MA, USA) A qPCR analysis was performed with the LightCycler 480 SYBR Green I Master mix (Roche, Ref. 04707516001) following the manufacturer’s instructions on the Lightcycler 480 instrument II system (Roche, Basel, Switzerland). The three (YWHAZ, TBP, PLOD1) out of six most stable reference genes were selected with the geNorm method of Vandesompele et al. [51] and the normalization factor was determined by the geometric mean. All primers for the qPCR reactions can be found in Appendix A. The relative expression (compared to the untreated control) of each gene was calculated using the (primer efficiency)^ΔCt^ method. 

## Figures and Tables

**Figure 1 ijms-21-07741-f001:**
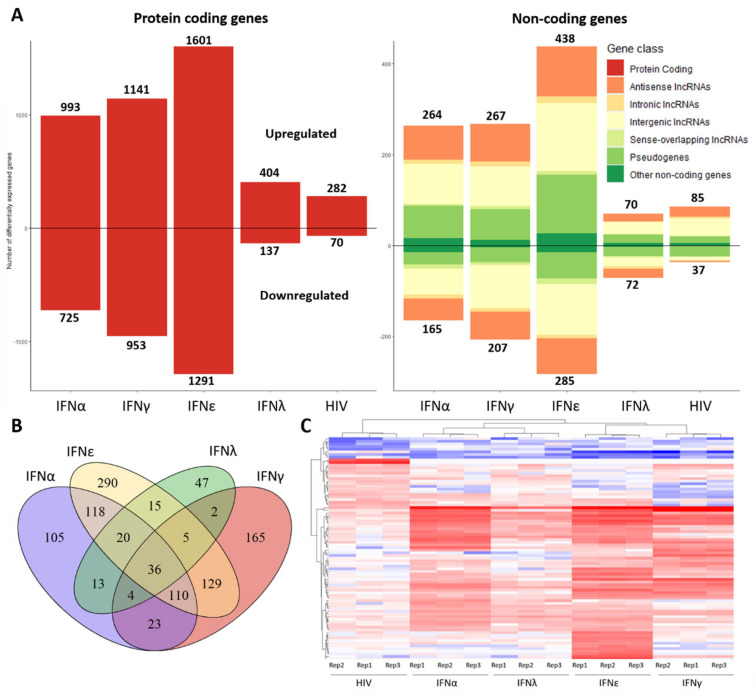
The differential expression profile of macrophages stimulated with IFN-α, IFN-ε, IFN-γ or IFN-λ or infected with HIV-1. (**A**) (Left) The number of differentially expressed protein coding genes (Padj < 0.05; log_2_ fold change > 1 or < −1). (Right) The differentially expressed non-coding genes. (**B**) Venn diagram showing the number of overlapping differentially expressed non-coding genes in the interferon stimulated conditions. (**C**) Heatmap including the top 25 differentially expressed non-coding genes for every condition were included. Color scale indicates the extend of the log_2_ fold change for each sample compared to the mean expression of the non-treated control (triplicate). Blue indicates downregulation (log_2_ FC < 0), red indicates upregulation (log_2_ FC < 0).

**Figure 2 ijms-21-07741-f002:**
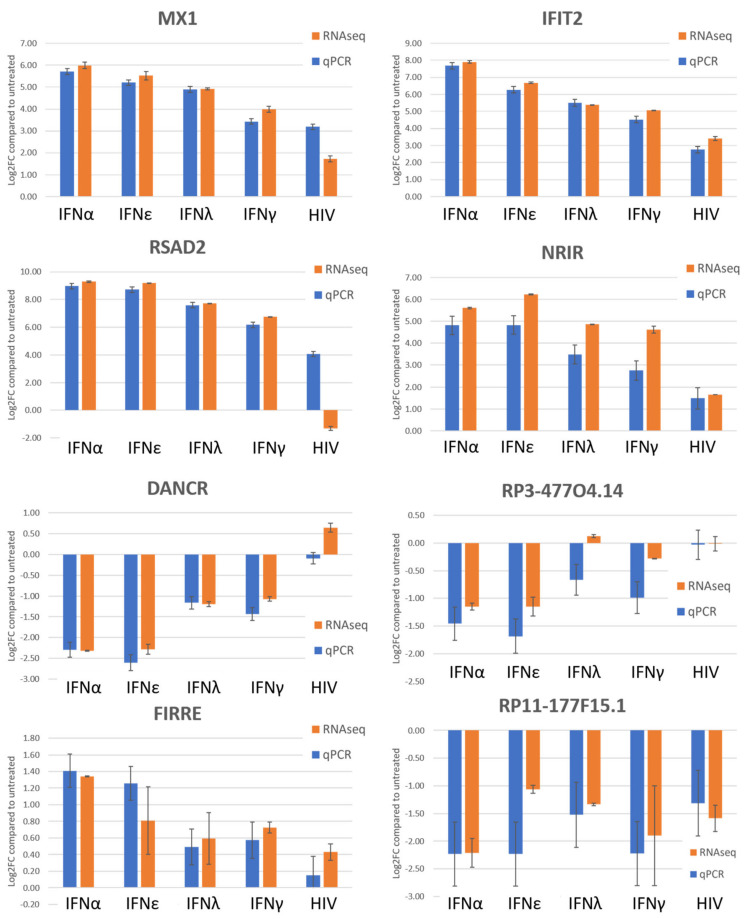
qPCR validation of three mRNAs and five non-coding genes. The RNA from the three replicates for every condition was pooled and the log_2_ fold change was determined compared to the untreated condition (n = 2). The data were normalized using three selected reference genes.

**Figure 3 ijms-21-07741-f003:**
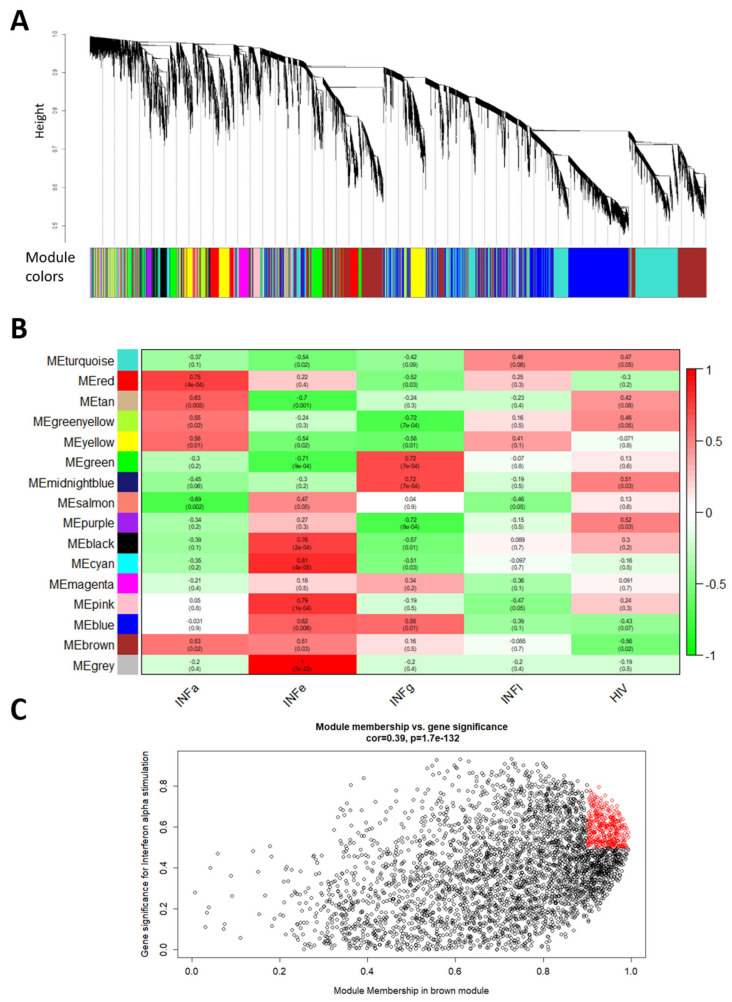
Weighted gene co-expression network analysis (WGCNA). (**A**) Hierarchical dendrogram of an unsigned, single-block gene co-expression network created based on bi-weight mid-correlation between gene expression over all samples. (**B**) Module–trait heatmap displaying the correlation between the eigengene of a module and the trait status (IFN-α, IFN-ε, IFN-γ, IFN-λ or HIV-1). Red indicates a positive correlation, while green indicates a negative correlation. (**C**) Scatterplot showing for each individual gene within the brown module the gene significance for IFN-α stimulation vs. its module membership. Genes that are considered hub genes are depicted in red.

**Figure 4 ijms-21-07741-f004:**
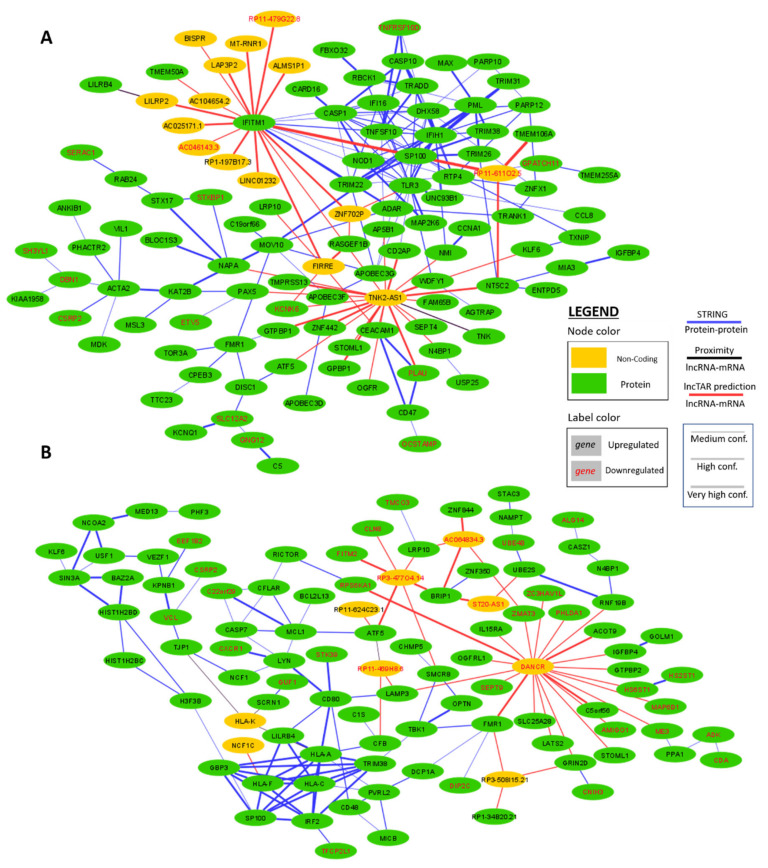
Networks displaying the predicted interactions between the differentially expressed (hub) genes of the brown module for: (**A**) IFN-α stimulation; and (**B**) IFN-ε stimulation. Three interaction types are depicted: (1) protein–protein interactions determined with the STRING webtool [24]; (2) Poly-A+ ncRNA–mRNA interactions predicted by LncTAR based on the binding free energy between the mRNA and the lncRNA sequences [25]; and (3) ncRNA–mRNA interactions based on genomic co-localization [26].

**Figure 5 ijms-21-07741-f005:**
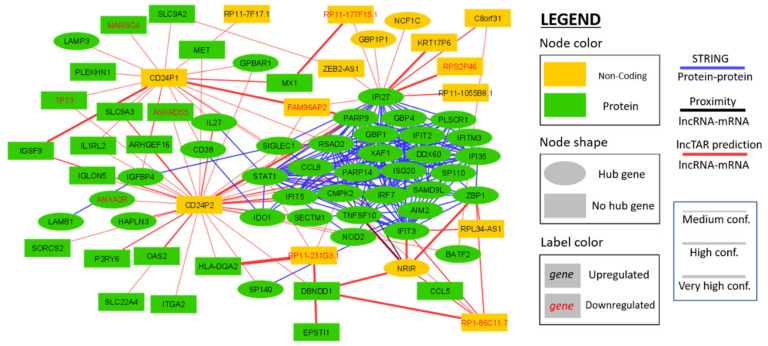
Network displaying the predicted interactions between the genes of the brown module that are differentially expressed upon HIV-1 infection. Three interaction types are depicted: (1) protein–protein interactions determined with the STRING webtool [24]; (2) Poly-A+ ncRNA–mRNA interactions predicted by LncTAR based on the binding free energy between the mRNA and the lncRNA sequences [25]; and (3) ncRNA–mRNA interactions based on genomic co-localization [26].

**Table 1 ijms-21-07741-t001:** Significantly enriched Gene Ontology Biological Process (GO_BP) terms for each module of the WGCNA. Modules not mentioned have no significant enriched term based on the Bonferroni-adjusted *p*-value.

Module	ID	Biological Process	Number of Genes (%)	Padj
Black	GO:0030198	extracellular matrix organization	27 (6.8)	1.32 × 10^−4^
Black	GO:0006954	inflammatory response	39 (4.8)	1.06 × 10^−3^
Black	GO:0030335	positive regulation of cell migration	32 (5.2)	3.15 × 10^−3^
Black	GO:0097529	myeloid leukocyte migration	18 (7.7)	5.02 × 10^−3^
Black	GO:0043062	extracellular structure organization	28 (5.5)	5.61 × 10^−3^
Blue	GO:0035966	response to topologically incorrect protein	66 (32.2)	4.91 × 10^−7^
Blue	GO:0070925	organelle assembly	205 (21.9)	6.09 × 10^−7^
Blue	GO:0034976	response to endoplasmic reticulum stress	82 (27.9)	7.73 × 10^−6^
Blue	GO:0007049	cell cycle	373 (18.9)	1.17 × 10^−5^
Blue	GO:0006974	cellular response to DNA damage stimulus	189 (21.4)	3.19 × 10^−5^
Brown	GO:0016032	viral process	213 (25.0)	2.61 × 10^−15^
Brown	GO:0044403	symbiotic process	222 (24.4)	9.79 × 10^−15^
Brown	GO:0044419	interspecies interaction between organisms	229 (23.9)	4.11 × 10^−14^
Brown	GO:0034341	response to interferon-gamma	69 (33.2)	6.15 × 10^−9^
Brown	GO:0045087	innate immune response	224 (21.4)	1.93 × 10^−8^
Cyan	GO:0098740	multi organism cell adhesion	2 (100)	2.98 × 10^−2^
Midnightblue	GO:0003009	skeletal muscle contraction	4 (8.3)	3.46 × 10^−2^
Pink	GO:0006119	oxidative phosphorylation	24 (15.7)	6.17 × 10^−15^
Pink	GO:0015985	energy coupled proton transport, down electrochemical gradient	24 (15.4)	9.85 × 10^−15^
Pink	GO:0015986	ATP synthesis coupled proton transport	24 (15.4)	9.85 × 10^−15^
Pink	GO:0009206	purine ribonucleoside triphosphate biosynthetic process	25 (13.0)	1.31 × 10^−13^
Pink	GO:0009145	purine nucleoside triphosphate biosynthetic process	25 (12.9)	1.49 × 10^−13^
Tan	GO:0050859	negative regulation of B cell receptor signaling pathway	3 (37.5)	2.22 × 10^−2^
Turquoise	GO:0007049	cell cycle	509 (25.8)	7.43 × 10^−23^
Turquoise	GO:0022402	cell cycle process	438 (26.8)	1.11 × 10^−22^
Turquoise	GO:0000278	mitotic cell cycle	316 (29.2)	1.58 × 10^−21^
Turquoise	GO:1903047	mitotic cell cycle process	316 (29.2)	1.58 × 10^−21^
Turquoise	GO:0140014	mitotic nuclear division	316 (29.2)	1.58 × 10^−21^
Yellow	GO:0042775	mitochondrial ATP synthesis coupled electron transport	19 (19.0)	2.99 × 10^−3^
Yellow	GO:0042773	ATP synthesis coupled electron transport	19 (18.8)	3.51 × 10^−3^
Yellow	GO:0090407	organophosphate biosynthetic process	68 (9.1)	8.61 × 10^−3^
Yellow	GO:0022900	electron transport chain	27 (13.4)	2.00 × 10^−2^
Yellow	GO:0019637	organophosphate metabolic process	95 (79.2)	3.04 × 10^−2^

**Table 2 ijms-21-07741-t002:** Significantly enriched pathways (KEGG/REACTOME) for each module of the WGCNA. Modules not mentioned have no significant enriched term based on the Bonferroni-adjusted *p*-value.

Module	ID	Biological Process	Source	Number of Genes (%)	Padj
Blue	1269649	Gene expression	REACTOME	372 (20.2)	1.63 × 10^−8^
Blue	1270038	Regulation of cholesterol biosynthesis by SREBP	REACTOME	25 (45.5)	1.01 × 10^−4^
Blue	1270039	Activation of gene expression by SREBF	REACTOME	21 (50.0)	1.51 × 10^−4^
Blue	1270037	Cholesterol biosynthesis	REACTOME	15 (62.5)	2.26 × 10^−4^
Blue	1268838	Organelle biogenesis and maintenance	REACTOME	86 (25.2)	4.61 × 10^−4^
Black	1270244	Extracellular matrix organization	REACTOME	24 (8.1)	1.85 × 10^−5^
Black	1470923	Interleukin-4 and 13 signaling	REACTOME	12 (10.2)	5.05 × 10^−3^
Black	1270254	Non-integrin membrane-ECM interactions	REACTOME	7 (15.2)	4.68 × 10^−2^
Brown	1269311	Interferon signaling	REACTOME	65 (32.2)	2.24 × 10^−7^
Brown	377873	Herpes simplex infection	KEGG	57 (30.8)	1.85 × 10^−5^
Brown	1269171	Adaptive immune system	REACTOME	175 (21.2)	9.65 × 10^−5^
Brown	1269312	Interferon alpha/beta signaling	REACTOME	28 (40.6)	2.02 × 10^−4^
Brown	1269314	Interferon gamma signaling	REACTOME	34 (36.2)	2.63 × 10^−4^
Pink	82942	Oxidative phosphorylation	KEGG	22 (16.5)	8.76 × 10^−14^
Pink	1270121	TCA cycle and respitory electron transport	REACTOME	24 (14.0)	1.64 × 10^−13^
Pink	83098	Parkinson’s disease	KEGG	21 (14.8)	4.69 × 10^−12^
Pink	1270127	Respiratory electron transport	REACTOME	20 (15.9)	5.28 × 10^−12^
Pink	83097	Alzheimer’s disease	KEGG	21 (12.3)	2.06 × 10^−10^
Tan	1427857	Regulation of TLR by endogeneous ligand	REACTOME	3 (18.3)	3.21 × 10^−2^
Turquoise	1269741	Cell cycle	REACTOME	215 (34.5)	7.99 × 10^−21^
Turquoise	1269763	Cell cycle, mitotic	REACTOME	183 (35.4)	8.48 × 10^−19^
Turquoise	1427846	rRNA processing in the nucleus and cytosol	REACTOME	87 (45.1)	4.25 × 10^−15^
Turquoise	1383086	Major pathway of rRNA processing	REACTOME	83 (45.9)	4.25 × 10^−15^
Turquoise	1269649	Gene expression	REACTOME	468 (25.4)	9.78 × 10^−15^
Yellow	1270128	Respiratory electron transport	REACTOME	18 (17.5)	1.42 × 10^−2^
Yellow	82942	Oxidative phosphorylation	KEGG	20 (15.0)	4.66 × 10^−2^

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
