# Peer review of "Interferon-Mediated Long Non-Coding RNA Response in Macrophages in the Context of HIV"

_ijms, 2020, doi:10.3390/ijms21207741_

Round 1

Reviewer 1 Report

The manuscript by Schynkel and colleagues details a study of the pattern of expression of long non-coding RNAs in primary human macrophages in response to stimulation with multiple IFNs and after 6 hours of infection with HIV. The authors describe the changes in the pattern of expression of both lncRNAs and protein-coding RNAs, and compare and contrast the response to the tested stimuli. Finally, they attempt to generate a framework for the observed changes by performing WGCNA study and use two sets of data as additional constraints: the ability of lncRNAs to basepair to protein-coding RNAs, and the proximity of genomic loci of lncRNAs and protein-coding RNAs. While this is certainly a worthy study and of use to the field, there are important issues that have to be solved. Importantly, the authors do not need to traverse to shaky scientific foundation in an attempt to make sense of their data which results in misleading outcomes. The study, in the absence of such scientifically unjustified additions, is already worthy of publication. The issues are described below:

Major issues:

1) The authors are using binding energies based on the in silico calculated ability of a lncRNA to binding an mRNA as a constraint in their network development study. This approach is erroneous and lacks scientific basis. To date only a small fraction of lncRNAs are thought to function by basepairing to an mRNA. Since this mechanism of function is not relevant to the vast majority of lncRNAs, basepairing ability must not be used as a global criteria in network model development. Further, in co-expression networks, there is no way to distinguish direct versus indirect impacts, so assuming that there should be a contact of any sort (basepairing or else) between the identified co-expressed molecules is not scientifically supported. Such approaches can only be applied to specific lncRNAs for which such mechanisms of function have been shown, such as ZEB2-AS1 (which is one of the lncRNAs the authors have identified). In the absence of prior knowledge about the mode of function of a lncRNA, the use of such data will only create misleading and scientifically invalid data. Evidence for the lack of validity of this approach is that in the case of studied lncRNAs (e.g. lncRNA NRIR that the authors prominently feature), the known targets of the lncRNA have not been captured by the method used by the authors. This type of scientifically unjustified approaches do more harm than good, and in addition to generating misleading and biologically irrelevant data, create a precedent for more misleading experimentation by others. The authors must completely remove these basepairing data from all except the lncRNAs known to function through basepairing to their targets. Importantly, the rest of their data are solid, and it’s not clear why the authors weaken and diminish the value of their data by including such scientifically unjustified information.

2) As mentioned above, a number of the lncRNAs identified by the authors have known targets. It’s surprising that the authors are not using this information in developing their network models. If the goal is anything other than merely a computational biology exercise, such data must be included to generate a network model that can be used by the rest of the field and is based on experimentally validated data.

Minor issues:

There are many factually incorrect and misleading statements in the manuscript that have to be corrected.

1) Line 49: Not necessarily true. There has been reports in literatuare that some lncRNAs originate from pol I and pol III transcriptional activity. This sentence needs to be corrected..

2) line 50 and 51: Not necessarily. Two of the most abundantly transcribe cellular RNAs are MALAT1 and NEAT1, both lncRNAs. Anyone who has done single cell RNA-seq knows that there are many examples of very highly expressed lncRNAs. Also, the ultra-conserved regions of the human genome give rise to lncRNAs, in a number of studied cases. So some lncRNAs are among the most conserved cellular RNAs. The authors need to refrain from generalization and carefully qualify their sentences.

For example:

A significant fraction of lncRNAs are less conserved than protein-coding genes and are also expressed at lower levels compared to protein-coding genes...

2) Lines 85 and 86: the numbers cited in the text do not match those in figure 1A.

3) Table S1 and S2 need more details in the legend. In the column labeled "percentage of genes in annotation", two numbers, separated by a comma are shown. Similarly, in the FDR and Adjusted p-value columns. Are the authors using commas instead of periods? This should be corrected!

4) line 214: this definition of lncRNAs is already outdated, as there are lncRNAs that are known to code for small peptides, however, they also have a distinct main function that is directed by the RNA itself. The sentence in this line should be changed to reflect this new trend and understanding in the field.

5) Line 215: The authors indicate that the majority of the non-coding RNAs identified in this study are functional lncRNAs, and the rest are expressed pseudogenes. What do the authors mean by "functional"? In the field of lncRNAs, functional is reserved for a lncRNA that has been studied and is known to have a cellular function. The authors don't seem to be using this nomenclature. Especially that there are pseudogenes that are considered functional lncRNAs, as they have been studied and an RNA-mediated function has been identified for them. This misleading sentence must be corrected.

6) The authors use the term “non-coding” (as opposed to long non-coding RNA) throughout the manuscript. Since the authors are not studying short ncRNAs, they should refrain from the use of "non-coding" which is misleading and creates the expectation that miRNAs or other small RNAs are also being studied. Especially that the study is limited to polyA+ lncRNAs only. This information should be included in the abstract and throughout the manuscript to ensure the readers know which subclass of lncRNAs is being characterized in this study.

7) The authors need to use a uniform gene name scheme. In one part of the manuscript, they use the official gene symbol RSAD2, while elsewhere they use viperin. In both cases, it's best to use both gene names, as they are both in use, e.g. RSAD2/viperin.

8) Line 266-267: this is a factual error. In the cited reference, BST2 is regulated by its bidirectional promoter-sharing lncRNA BISPR, which is also represented in the lncRNAs identified in this study (and included in the drawn networks).

9) Line272: it is appropriate to discuss the fact that lncRNA IR3945HG is a miRNA host and thus, any observed effects can be due to changes in the level of the miRNA, not the lncRNA host.

Author Response

The manuscript by Schynkel and colleagues details a study of the pattern of expression of long non-coding RNAs in primary human macrophages in response to stimulation with multiple IFNs and after 6 hours of infection with HIV. The authors describe the changes in the pattern of expression of both lncRNAs and protein-coding RNAs, and compare and contrast the response to the tested stimuli. Finally, they attempt to generate a framework for the observed changes by performing WGCNA study and use two sets of data as additional constraints: the ability of lncRNAs to basepair to protein-coding RNAs, and the proximity of genomic loci of lncRNAs and protein-coding RNAs. While this is certainly a worthy study and of use to the field, there are important issues that have to be solved. Importantly, the authors do not need to traverse to shaky scientific foundation in an attempt to make sense of their data which results in misleading outcomes. The study, in the absence of such scientifically unjustified additions, is already worthy of publication. The issues are described below:

Major issues:

1) The authors are using binding energies based on the in silico calculated ability of a lncRNA to binding an mRNA as a constraint in their network development study. This approach is erroneous and lacks scientific basis. To date only a small fraction of lncRNAs are thought to function by basepairing to an mRNA. Since this mechanism of function is not relevant to the vast majority of lncRNAs, basepairing ability must not be used as a global criteria in network model development. Further, in co-expression networks, there is no way to distinguish direct versus indirect impacts, so assuming that there should be a contact of any sort (basepairing or else) between the identified co-expressed molecules is not scientifically supported. Such approaches can only be applied to specific lncRNAs for which such mechanisms of function have been shown, such as ZEB2-AS1 (which is one of the lncRNAs the authors have identified). In the absence of prior knowledge about the mode of function of a lncRNA, the use of such data will only create misleading and scientifically invalid data. Evidence for the lack of validity of this approach is that in the case of studied lncRNAs (e.g. lncRNA NRIR that the authors prominently feature), the known targets of the lncRNA have not been captured by the method used by the authors. This type of scientifically unjustified approaches do more harm than good, and in addition to generating misleading and biologically irrelevant data, create a precedent for more misleading experimentation by others. The authors must completely remove these basepairing data from all except the lncRNAs known to function through basepairing to their targets. Importantly, the rest of their data are solid, and it’s not clear why the authors weaken and diminish the value of their data by including such scientifically unjustified information.

Thank you for your elaborate comment on the use of the lncTAR tool to predict lncRNA-mRNA interactions that were included in the lncRNA-mRNA-protein interaction networks. We acknowledge that today a rather small number of lncRNA-mRNA interactions have been described and validated, which is part of the reason that we as a lncRNA research community resort to these tools. This also indicates that direct lncRNA-mRNA interactions do exist, with functions ranging from modulation of mRNA splicing, to repression/activation of mRNA translation, or protection from mRNA decay (Review Yoon et al., 2012, J Mol Biol and review Dykes and Emanueli, 2017, GPB). We agree that constructing a network with only experimentally validated interactions would reduce the noise in our network but on the other hand would remove the aspect of exploring possible new interactions and prioritizing potential hub lncRNA for in vitro validation. Furthermore, we decided on a conservative use of lncTAR and only included lncRNA-mRNA interactions which corresponds to a prediction with high confidence (normalized binding free energy < -0.15). The lncRNAs highlighted in the discussion are those with multiple of those high confidence predicted lncRNA-mRNA interactions, which at least indicates a possible binding capacity. Overall and even without the lncTAR predictions, we are aware that the biological impact of these hub lncRNAs will have to be investigated in follow-up studies.

To make sure that the reader understands this message more clearly and that the depicted lncRNA-mRNA interactions are predictions we have adjusted the manuscript:

  1. By adding “predicted” in the legend of each network,
  2. Changing the caption of the figures to “poly-A+ ncRNA-mRNA interactions predicted by LncTAR”,
  3. In the text of the results (part 2.4) we added: “The network consist of protein-protein interactions, and lncRNA-mRNA interactions that were predicted based on the genomic location of the genes and the normalized binding free energy between the lncRNA and mRNA sequence.”
  4. In the discussion we added (line 293-294): “We want to note that the lncRNA-mRNA interactions are merely predictions and future in vitro validation has to assess actual lncRNA-mRNA binding.”

2) As mentioned above, a number of the lncRNAs identified by the authors have known targets. It’s surprising that the authors are not using this information in developing their network models. If the goal is anything other than merely a computational biology exercise, such data must be included to generate a network model that can be used by the rest of the field and is based on experimentally validated data.

It is indeed correct that a number of lncRNAs have known targets that are not depicted in the networks. The lncRNA ZEB2-AS1 for example, like mentioned by reviewer 1, activates the transcription of the protein coding gene ZEB2 by modulating its transcription. However, the WGCNA has classified ZEB2 in the blue module, therefore the interaction does not show up in the interferon-related network constructed with the brown module that was selected and assessed in more detail based on the gene enrichment results. This is inherent to a WGCNA that classifies genes based on correlated RNA expression profiles. In our opinion, we have to let the data speak, rather than adding known interactions that might be present in certain conditions but maybe not during IFN induction or HIV infection. It is worth mentioning however that we did also pick up known interaction (also described in the discussion) like the NRIR-CMPK2 or NRIR-RSAD2/viperin interaction, validating this WGCNA methodology.

Minor issues:

There are many factually incorrect and misleading statements in the manuscript that have to be corrected.

1) Line 49: Not necessarily true. There has been reports in literatuare that some lncRNAs originate from pol I and pol III transcriptional activity. This sentence needs to be corrected..

The sentence was corrected to “their transcription depends most often on RNA polymerase II activity”. 

2) line 50 and 51: Not necessarily. Two of the most abundantly transcribe cellular RNAs are MALAT1 and NEAT1, both lncRNAs. Anyone who has done single cell RNA-seq knows that there are many examples of very highly expressed lncRNAs. Also, the ultra-conserved regions of the human genome give rise to lncRNAs, in a number of studied cases. So some lncRNAs are among the most conserved cellular RNAs. The authors need to refrain from generalization and carefully qualify their sentences.

For example:

A significant fraction of lncRNAs are less conserved than protein-coding genes and are also expressed at lower levels compared to protein-coding genes...

Thank you for your suggestion, this sentence has been incorporated in the introduction.

2) Lines 85 and 86: the numbers cited in the text do not match those in figure 1A.

We have crosschecked the numbers cited in the text and figure 1A. The text is as following : “Upon HIV-1 infection a total of 474 genes were differentially expressed, of which 352 were mRNAs, 38 pseudogenes and 78 lncRNAs (figure 1A).”. In figure 1A you see for the HIV infected condition a total of 474 (282 + 70 + 85 + 37) DE genes. These include amongst others 352 (282+70) mRNAs.

3) Table S1 and S2 need more details in the legend. In the column labeled "percentage of genes in annotation", two numbers, separated by a comma are shown. Similarly, in the FDR and Adjusted p-value columns. Are the authors using commas instead of periods? This should be corrected!

The numbers were indeed written in ‘French’ notation. We have adapted this to the notation with periods instead of commas both in tables in the main text, as in the supplemental.

4) line 214: this definition of lncRNAs is already outdated, as there are lncRNAs that are known to code for small peptides, however, they also have a distinct main function that is directed by the RNA itself. The sentence in this line should be changed to reflect this new trend and understanding in the field.

Sentence was adapted to: “So their main function does not depend on mRNA to protein translation, but it is rather RNA transcription itself or the RNA molecule that fulfills the purpose of the gene.” 

5) Line 215: The authors indicate that the majority of the non-coding RNAs identified in this study are functional lncRNAs, and the rest are expressed pseudogenes. What do the authors mean by "functional"? In the field of lncRNAs, functional is reserved for a lncRNA that has been studied and is known to have a cellular function. The authors don't seem to be using this nomenclature. Especially that there are pseudogenes that are considered functional lncRNAs, as they have been studied and an RNA-mediated function has been identified for them. This misleading sentence must be corrected.

Indeed, there is a rather artificial line between lncRNAs and pseudogenes, as a pseudogene could also be considered a lncRNA when its RNA product has any function. The word “functional” was left out to exclude any unintended confusion or misleading interpretation.

6) The authors use the term “non-coding” (as opposed to long non-coding RNA) throughout the manuscript. Since the authors are not studying short ncRNAs, they should refrain from the use of "non-coding" which is misleading and creates the expectation that miRNAs or other small RNAs are also being studied. Especially that the study is limited to polyA+ lncRNAs only. This information should be included in the abstract and throughout the manuscript to ensure the readers know which subclass of lncRNAs is being characterized in this study.

Both in the abstract as in the main text, the term ncRNAs was adapted to poly-A+ ncRNA to differentiate with other ncRNAs families, i.e. miRNAs.

7) The authors need to use a uniform gene name scheme. In one part of the manuscript, they use the official gene symbol RSAD2, while elsewhere they use viperin. In both cases, it's best to use both gene names, as they are both in use, e.g. RSAD2/viperin.

The gene name was consistently adapted to RSAD2/viperin.

8) Line 266-267: this is a factual error. In the cited reference, BST2 is regulated by its bidirectional promoter-sharing lncRNA BISPR, which is also represented in the lncRNAs identified in this study (and included in the drawn networks).

This was indeed incorrect and has been removed from the discussion.

9) Line272: it is appropriate to discuss the fact that lncRNA IR3945HG is a miRNA host and thus, any observed effects can be due to changes in the level of the miRNA, not the lncRNA host.

This was added to the discussion: “Transcription of MIR3945HG might also regulate the expression of its hosting miRNA, rather than function on its self.”

Reviewer 2 Report

In this manuscript, Schynkel et al conducted a comprehensive analysis of IFN-induced long noncoding RNA (lncRNA) in macrophages and in macrophages infected with HIV-1. The authors also performed correlation analysis of mRNA and lncRNA to reveal potential importance of these identified lncRNA. This well-written manuscript provides a nicely performed analysis focusing on identification of IFN related lncRNA.

There are only a few very minor points needed further explanation and clarification.

  1. It seems that the previous study (Szaniawski et al, mBio 2018) cited in this manuscript did not deposit their RNA-Seq data in public database; however, their sequencing data were further analyzed in this study. It might be better to make it clear in the first paragraph of the Result section.
  2. The discrepancy between RNA-seq analysis and quantitative PCR analysis also raised some concerns about the relibility of some of the lncRNA analysis in this study. The authors attributed this discranpcy to “general low read counts of lncRNA”. Moreover, the RNA-seq library from previous study was generated through poly-A purified methods, which for sure will miss lots of lncRNAs. These “defects” might further questionate the utilization of the RNA-seq data from previous study for lncRNA analysis.

Author Response

In this manuscript, Schynkel et al conducted a comprehensive analysis of IFN-induced long noncoding RNA (lncRNA) in macrophages and in macrophages infected with HIV-1. The authors also performed correlation analysis of mRNA and lncRNA to reveal potential importance of these identified lncRNA. This well-written manuscript provides a nicely performed analysis focusing on identification of IFN related lncRNA.

There are only a few very minor points needed further explanation and clarification.

  1. It seems that the previous study (Szaniawski et al, mBio 2018) cited in this manuscript did not deposit their RNA-Seq data in public database; however, their sequencing data were further analyzed in this study. It might be better to make it clear in the first paragraph of the Result section.

The data was not publicly available at the time of submission. In the meantime this was finalized and the GEO accession number was added to the manuscript. From October 2nd 2020 on the data will be available for everyone (for now a token is needed to access the data). We have clarified this in the beginning of the result section by providing the accession number.

  1. The discrepancy between RNA-seq analysis and quantitative PCR analysis also raised some concerns about the relibility of some of the lncRNA analysis in this study. The authors attributed this discranpcy to “general low read counts of lncRNA”. Moreover, the RNA-seq library from previous study was generated through poly-A purified methods, which for sure will miss lots of lncRNAs. These “defects” might further questionate the utilization of the RNA-seq data from previous study for lncRNA analysis.

We acknowledge that poly-A purification of the RNA library conceals about half of the total lncRNA response to the IFN induction or HIV infection. However, in our opinion, this does not reduce the significance of the identified poly-A tailed lncRNAs. These are indeed differentially expressed upon IFN induction or HIV infection and thus biologically relevant and worth investigating. A total RNA library would have identified more lncRNA hub genes, but would not have exclude the currently described poly-A tailed hub lncRNAs. To make this clear, we have added the information of poly-A purification to the abstract and changed ‘ncRNAs’ to ‘poly-A+ ncRNAs’ throughout the main text, to notify the reader on this fact.

Concerning the discrepancies between in the RNAseq and the qPCR analysis. We would like to highlight that only three out of all 40 tested genes/condition show a divergent log2 fold change. In 93% of the conditions the qPCR results are in line with the RNAseq results, which is in line with previous reported studies which include qPCR validation of RNAseq results. Therefore, we consider the RNAseq data as trustworthy.

In an effort to explain the three unvalidated genes we agree that ‘general low read counts of lncRNA’ does not reflect the total story, as we have included adequate QC parameters to investigate lncRNAs with sufficient high coverage. We also mention “the different way of normalization, multiple splicing variant detection and the pooling of the RNAseq triplicates for qPCR analysis may explain these few discrepancies”. Indeed, the pooling of the RNAseq triplicates can influence the qPCR outcome, as a single high outlier can conceal the expression level in the other two replicates. Ideally, we would have done the qPCR on all three replicates separately, but these RNA samples were unfortunately only available as pooled.

Reviewer 3 Report

Based on previously published RNA-seq data, the authors analyzed the patterns of lncRNAs expression in MDM stimulated with different types of IFNs or infected with HIV, which revealed an extensive lncRNA response by IFNs and HIV. The analysis also identified "hub" lncRNAs within IFN- or HIV-associated gene clusters. This is a "data-mining" paper and there are no solid biological conclusions supported by experimental data. However, it provides important information that helps to elucidate possible roles of lncRNA in IFN-stimulated or HIV-infected cells, which could be further examined as future studies. Several points should be addressed, however, for a publication in VIRUSES.

Specific Points

  1. As far as I understand, all bioinformatic data shown in this manuscript were based on the RNA-seq data from the previous publication, and there are no new RNA-seq data. If so, it is not appropriate to have a section of RNA-seq in the Materials and Methods.
  2. There are several specific lncRNAs reported to be associated with HIV (for example, Li et al Nat. Comm. 2016, PMID: 27291871). Are these lncRNA up- or down-regulated by IFN or HIV?
  3. Since HIV-based lentiviral vector are popularly used to expressed genes in primary cells, it is of interest/importance to examine whether VSV-G-pseudotyped lentivirus also induces similar sets of lncRNAs.
  4. (Fig.2) RNA-seq data and RTqPCR data agree very well with each other for all IFNs, but not HIV, which is very interesting. Author should describe more about this. For example, are there any differences in conditions of HIV infection between these two experiments (MOIs, virus prep, percentage of HIV infected cells, etc)?
  5. Similarly, it is of interest to examine whether HIV infection induces specific IFN production (RTqPCR or ELISA).
  6. Page 9, line 276. "The RNA-seq data was" should be "The RNA-seq data were".

Author Response

Based on previously published RNA-seq data, the authors analyzed the patterns of lncRNAs expression in MDM stimulated with different types of IFNs or infected with HIV, which revealed an extensive lncRNA response by IFNs and HIV. The analysis also identified "hub" lncRNAs within IFN- or HIV-associated gene clusters. This is a "data-mining" paper and there are no solid biological conclusions supported by experimental data. However, it provides important information that helps to elucidate possible roles of lncRNA in IFN-stimulated or HIV-infected cells, which could be further examined as future studies. Several points should be addressed, however, for a publication in VIRUSES.

Specific Points

  1. As far as I understand, all bioinformatic data shown in this manuscript were based on the RNA-seq data from the previous publication, and there are no new RNA-seq data. If so, it is not appropriate to have a section of RNA-seq in the Materials and Methods.

The RNA-seq data originates from a previous publication of Szaniawski et al. and we follow the reviewers suggestion. However, we feel like it would be worthwhile to give the reader limited background information which lies at the basis of this manuscript and helps the reader to frame the analysis performed, i.e. the poly-A purification during the RNAseq library preparation. Therefore, we heavily shortened the length of the material and methods section concerning the generation of the data and clarified the origin of the RNAseq dataset. Furthermore the accession number of the publicly available data is now provided.

  1. There are several specific lncRNAs reported to be associated with HIV (for example, Li et al Nat. Comm. 2016, PMID: 27291871). Are these lncRNA up- or down-regulated by IFN or HIV?

The lncRNA mentioned by the reviewer, NRON, was not picked up in any of the RNAseq samples, most likely because NRON has no poly-A tail and will have been lost in the poly-A purification step during the library preparation.  Other (host) lncRNAs previously associated with HIV are NEAT1, HEAL, GAS5 and MALAT1. They were upregulated but not significant (log2FC < 1), with HEAL the highest log2FC of 0.61, this was included in the results: “LncRNAs NEAT1, HEAL, GAS5 and MALAT1 have all been previously reported to be associated with HIV infection, but we observed no significant differential expression in HIV-1 infected MDMs.”. Therefore, these lncRNAs were not included in the results and discussion.

  1. Since HIV-based lentiviral vector are popularly used to expressed genes in primary cells, it is of interest/importance to examine whether VSV-G-pseudotyped lentivirus also induces similar sets of lncRNAs.

Indeed, it is likely that VSV-G pseudotyped lentiviral expression vectors will induce at least some type of transcriptional response (i.e. antiviral and/or IFN-mediated) which also will affect protein-coding genes. This indeed could be an additional factor to account for in transcriptional analysis in those kinds of experiments and although it would be interesting to investigate this in the future, this is at this point beyond the scope of the current article.

  1. (Fig.2) RNA-seq data and RTqPCR data agree very well with each other for all IFNs, but not HIV, which is very interesting. Author should describe more about this. For example, are there any differences in conditions of HIV infection between these two experiments (MOIs, virus prep, percentage of HIV infected cells, etc)?

The RNAseq analysis and RTqPCR analysis have been performed on the same cDNA libraries, although for the RTqPCR analysis the three replicates were pooled. Therefore, there are no differences in conditions of HIV infection between the two analyses and the discrepancies can be purely contributed to differences between RNAseq and RTqPCR. In the manuscript we mention “the general low read counts of lncRNA” and “the different way of normalization, multiple splicing variant detection and the pooling of the RNAseq triplicates for qPCR analysis” as the possible explanation for the few discrepancies between the RTqPCR and the RNAseq analysis.

  1. Similarly, it is of interest to examine whether HIV infection induces specific IFN production (RTqPCR or ELISA).

It is known that macrophages are potent producers of IFNα (Cantell and Pirhonen, 1996, J Interferon Cytokine Res.) and they also have been shown to produce IFNγ upon stimulation with IL-12 and IL-18 (Darwich et al, 2009, Immunology), IL-23 (Hou et al., 2018, Protein Cell.) or Mycobacterium tuberculosis infection (Robinson et al, 2010, J Innate Immun). To what extend HIV infection induces several IFN in macrophages is not completely clear and indeed interesting. We looked at the RNA-seq data but could not pinpoint upregulation of specific IFNε, IFNα, IFNγ or IFNλ transcripts. However, we did find upregulation of IRF7 which could indicate type I IFN induction. Although this merit further examination in follow-up work, we would argue that the message, focus and conclusions in our manuscript hold without current further exploration.

  1. Page 9, line 276. "The RNA-seq data was" should be "The RNA-seq data were".

This was adapted.

Round 2

Reviewer 1 Report

The majority of the issues raised by this reviewer have been addressed in the revised version. Some issues remain, including the use of a scientifically unjustified constraint during network modeling. Unlike what the authors mention, unpublished work presented in meetings indicates that the mechanism of action of NRIR is not through forming basepairs with other RNAs, certainly not for RSAD2 and CMPK2. Also, the shape of network modeled around NRIR indicates the lack of scientific validity of this approach. In order to preserve scientific integrity, the authors need to add a sentence to the manuscript indicating that:

formation of lncRNA-mRNA basepairs is not considered a common functional mechanism for lncRNAs and thus, the derived network models, which partly rely on predicted lncRNA-mRNA interactions, need to be validated for accuracy.

Author Response

Line 290-292. The suggested statement of reviewer 1 was added to the discussion as following: "We want to note that the lncRNA-mRNA interactions are merely predictions that are in general not considered as a very common functional mechanism for lncRNAs. Therefore, future in vitro validation will have to assess actual lncRNA-mRNA binding and its functional impact."